# CountXplain: Interpretable Cell Counting with Prototype-Based Density Map Estimation

**Abdurahman Ali Mohammed** [1]               ABDU@IASTATE.EDU
**Wallapak Tavanapong** [1]                TAVANAPO@IASTATE.EDU
**Catherine Fonder** [2,3,5]                CFONDER@IASTATE.EDU
**Donald S. Sakaguchi** [2,3,4,5]              DSSAKAGU@IASTATE.EDU

[1] *Department of Computer Science, Iowa State University, Ames, IA 50011*

[2] *Department of Genetics, Development, and Cell Biology, Iowa State University, Ames, IA 50011*

[3] *Molecular, Cellular, and Developmental Biology Program, Iowa State University, Ames, IA 50011*

[4] *Neuroscience Program, Iowa State University, Ames, IA 50011*

[5] *Nanovaccine Institute, Iowa State University, Ames, IA 50011*

**Editors:** Accepted for publication at MIDL 2025

## Abstract

Cell counting in biomedical imaging is pivotal for various clinical applications, yet the interpretability of deep learning models in this domain remains a significant challenge. We propose a novel prototype-based method for interpretable cell counting via density map estimation. Our approach integrates a prototype layer into the density estimation network, enabling the model to learn representative visual patterns for both cells and background artifacts. The learned prototypes were evaluated through a survey of biologists, who confirmed the relevance of the visual patterns identified, further validating the interpretability of the model. By generating interpretations that highlight regions in the input image most similar to each prototype, our method offers a clear understanding of how the model identifies and counts cells. Extensive experiments on two public datasets demonstrate that our method achieves interpretability without compromising counting effectiveness. This work provides researchers and clinicians with a transparent and reliable tool for cell counting, potentially increasing trust and accelerating the adoption of deep learning in critical biomedical applications. Code is available at https://github.com/NRT-D4/CountXplain.
**Keywords:** Cell Counting, Biomedical Imaging, Deep Learning, Interpretability, Density Map Estimation

## 1. Introduction

Accurate cell counting is vital in biomedical imaging, enabling crucial insights into cellular processes. It is essential for disease diagnosis (Orth et al., 2017; Blumenreich, 1990), treatment evaluation (Polley et al., 2013), and biomedical research (Drost and Clevers, 2018; Das et al., 2017). Accurate counts are imperative for both scientific discovery and improving patient outcomes. Traditionally, cell counting relied on manual methods that are labor-intensive, prone to variability, and impractical for high-throughput tasks. Automated detection-based (segmentation and object detection) approaches aimed to localize and count individual cells (Arteta et al., 2012; Aldughayfiq et al., 2023; Morelli et al., 2021), but their performance degraded in densely packed or overlapping scenarios.

Density map estimation (DME) emerged as an effective alternative by predicting a density map where the sum over all the pixels in the map image corresponds to the object

count within the input image (Lempitsky and Zisserman, 2010). This approach excels in handling crowded and overlapping cells, making it particularly suited for biomedical applications with many cell counts in the range of hundreds to thousands per image. Deep learning methods, such as fully convolutional networks (Xie et al., 2018; Paul Cohen et al., 2017; Marsden et al., 2018; Zheng et al., 2024) and self-attention layers (Guo et al., 2019), have further enhanced DME's capabilities. Notably, CSRNet (Li et al., 2018), originally designed for crowd counting, was adapted for cell counting (Mumba Ngoyi et al., 2013; Mohammed et al., 2023), leveraging its ability to handle scale variation and complex spatial distributions.

In critical domains like healthcare and biomedicine, models that offer both accurate prediction and interpretability (the rationale behind model predictions) are highly desirable but are often difficult to achieve, especially for high-resolution images. Most interpretability techniques were proposed for classification tasks. Layer-wise Relevance Propagation (Bach et al., 2015), Class Activation Mapping (CAM) (Zhou et al., 2016), Grad-CAM (Selvaraju et al., 2017), and several others highlight regions influencing model predictions. Prototype-based methods like ProtoPNet (Chen et al., 2019), ProtoPShare (Rymarczyk et al., 2021), and ProtoVAE (Gautam et al., 2022) learn prototypes (i.e., generalized visual representation in latent space) for class-specific decisions, offering interpretable insights. ProtoPNet-based methods have been utilized in medical image classification tasks (Barnett et al., 2021; Mohammadjafari et al., 2021; Singh and Yow, 2021a,b; Djoumessi et al., 2024).

Interpretability methods for regression tasks on image input received much less attention. INSightR-Net (Hesse and Namburete, 2022) leverages a prototype layer to compare input images to learned prototypes, enabling intuitive visual explanations for diabetic retinopathy grading. Similarly, ExPeRT (Hesse et al., 2024) uses a prototype-based architecture for brain age prediction, incorporating optimal transport to align patches with learned prototypes. While effective for scalar predictions, the architectures of ExPeRT and Insight-RNet are not inherently designed for spatially distributed outputs like density maps. Both methods also rely on prototype labels, limiting applicability in scenarios like cell counting, where the number of cells in a single image in the same experiment can vary significantly from zero to thousands. *To date, we found no existing interpretability methods for density map estimation models.*

These limitations underscore the need for novel interpretability methods tailored to density map estimation, particularly in cell counting tasks. Addressing this gap could enable models to not only provide accurate predictions but also explain their reasoning in a way that is actionable for clinicians and researchers.

To bridge this gap, we propose CountXplain, a new prototype-based approach for interpretable cell counting via density map estimation. To our knowledge, this is the first framework integrating a prototype layer into the DME framework. Our method enables the model to learn prototypes, providing a transparent and intuitive explanation of its predictions without reducing its effectiveness in counting. These prototypes are used to generate interpretations that could help biologists understand the model's decisions for individual images as well as patterns the model has learned from the training set.

Our contributions are summarized as follows:

- CountXplain, a novel cell counting framework that integrates prototype learning with density map estimation, providing both accurate predictions and interpretability. The

framework utilizes two new components for the loss function: (1) *a prototype-to-feature loss* for getting cell prototypes to focus on regions containing cells and background prototypes to focus on background areas; and (2) *a diversity loss within each group of prototypes* to capture a wide range of feature patterns within each group.

- CountXplain performance on two public datasets achieves counting performance comparable to state-of-the-art models while providing interpretability.

- Our preliminary survey results with biologists demonstrate the relevance of the learned prototypes to cells and background artifacts.

CountXplain provides the accuracy of density map estimation and transparency in the model's decision-making, offering a valuable tool for biomedical research and applications.

## 2. Proposed Interpretable Prototype-based DME

Our design goal is to enable interpretability while maintaining accurate cell counting and spatial distribution of cells for each image. The proposed design using a new prototype layer can reveal patterns used in the model's decision for the predicted spatial distribution of cells per image and overall patterns the model has learned from the training set.

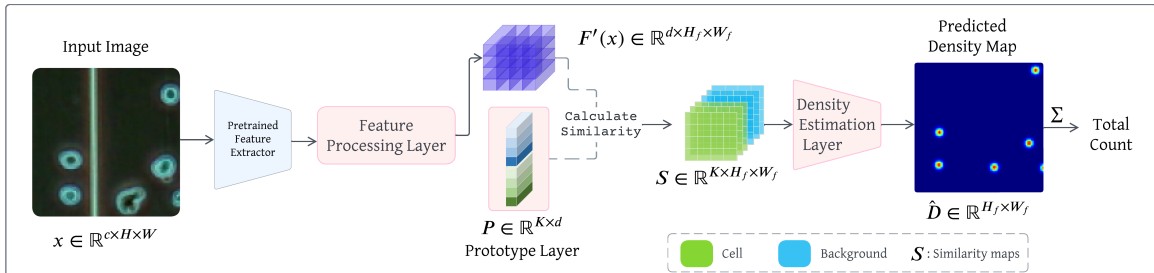

Figure 1: Architecture of CountXplain. See Section 2.1 for details.

### 2.1. Model Architecture

Our network architecture builds upon the feature extractor of a pre-trained density map estimation model based on a Convolutional Neural Network (CNN). We choose CNNs over Vision Transformers (ViTs) due to the limited availability of labeled data in this domain. CNNs leverage strong spatial inductive biases, making them more effective in data-scarce scenarios, whereas ViTs typically require large-scale datasets for optimal performance (Raghu et al., 2021; Park and Kim, 2022; D'Ascoli et al., 2021). Figure 1 illustrates the network architecture.

**Feature Extractor.** A pre-trained feature extractor $F(\mathbf{x})$ takes an input image $\mathbf{x} \in \mathbb{R}^{c \times H \times W}$ and produces a feature map $F(\mathbf{x}) \in \mathbb{R}^{d \times H_f \times W_f}$. The input image has $c$ channels, while the output has $d$ channels. $H$x$W$ and $H_f$x$W_f$ indicate the height and width per channel of the input image and the output density map, respectively.

**Feature Map Processing Layer.** To normalize the extracted features and prepare them for comparison with prototypes, we introduce a feature map processing layer. This layer

applies a 1x1 convolution followed by a sigmoid activation function, as shown in Eq. 1:

$$F'(\mathbf{x}) = \sigma(\text{Conv}_{1\times 1}(F(\mathbf{x}))), \tag{1}$$

where $\sigma$ denotes the sigmoid function, and $\text{Conv}_{1\times 1}$ represents a 1x1 convolutional operation. The output $F'(\mathbf{x}) \in \mathbb{R}^{d \times H_f \times W_f}$ has the same spatial dimensions as $F(\mathbf{x})$.

**Prototype Layer.** The prototype layer forms the core of our interpretable approach. It has a set of $K$ learnable prototypes $P = \{p_1, \ldots, p_K\}$, where each prototype $p_i \in \mathbb{R}^d$ is a vector in the same space as each spatial location in $F'(\mathbf{x})$. The $K$ prototypes are divided into $K_{cell}$ cell prototypes and $K_{bg}$ background prototypes. The cell prototypes capture features of cells to be counted, while the background prototypes focus on non-counted elements, such as background and other artifacts. We want the background prototypes to show the domain experts the kind of background and other artifacts the model recognizes.

For each spatial location $(h, w)$ in the processed feature map, we compute the squared L2 distance with each prototype. This operation results in a distance map $\Phi(\mathbf{x}) \in \mathbb{R}^{K \times H_f \times W_f}$. Following (Chen et al., 2019), we convert these distances to similarities using a log-based transformation, resulting in a similarity map $S(\mathbf{x}) \in \mathbb{R}^{K \times H_f \times W_f}$. Higher values in the similarity map indicate greater similarity between a prototype and the corresponding region in the feature map. See Figure 1.

**Density Estimation Layer.** The layer uses a single convolutional layer with a $1 \times 1$ kernel to transform the similarity map $S$ into the final 1-channel density map, denoted as $\hat{D}(\mathbf{x}) \in \mathbb{R}^{H_f \times W_f}$, as defined in Eq. 2.

$$\hat{D}(\mathbf{x}) = \sum_{i=1}^{K} \theta_i S_i(\mathbf{x}), \tag{2}$$

where $S_i(\mathbf{x})$ represents the similarity map corresponding to prototype $i$, and $\theta_i$ denotes the learnable weight associated with prototype $i$ in the $1 \times 1$ convolutional layer.

The use of a convolutional layer enables learning of the weight of each prototype's contribution to the final density estimate. This $1 \times 1$ convolution computes a learned linear combination of the prototype similarity maps, maintaining interpretability while allowing for more nuanced density estimation than simple averaging. The total cell count can then be obtained by summing over this density map as $\sum_{h=1}^{H_f} \sum_{w=1}^{W_f} \hat{D}_{h,w},(\mathbf{x})$ where $\hat{D}_{h,w}(\mathbf{x})$ is the predicted density at location $(h, w)$.

## 2.2. Training Procedure and Loss Function

Our training procedure aims to optimize the model's parameters, including the prototypes and the weights of the density estimation layer, while keeping the feature extractor frozen to preserve its pre-trained knowledge.

### 2.2.1. LOSS FUNCTION

We use a multi-objective loss function shown in Eq. 3 that balances accuracy and interpretability.

$$\mathcal{L}_{total} = \lambda_1 \mathcal{L}_{density} + \lambda_2 \mathcal{L}_{proto-feature} + \lambda_3 \mathcal{L}_{diversity}, \tag{3}$$

where $\lambda_1$, $\lambda_2$, and $\lambda_3$ are weighting coefficients that control the balance between the different objectives. We will now describe each component of this loss function in detail.

**Density Estimation Loss ($\mathcal{L}_{density}$)** The primary objective of our model is to accurately estimate the cell density, minimizing the Mean Squared Error between the predicted density map and the ground truth for all samples. This loss term ensures that our model learns to generate density maps that closely match the ground truth density maps across all samples in the training dataset.

**Proposed Prototype-to-Feature Loss ($\mathcal{L}_{proto-feature}$)** This loss term shown in Eq. 4 has two components: loss for cell prototypes ($\mathcal{L}_{cell}$) and loss for background prototypes ($\mathcal{L}_{bg}$) in Eq. 5. Recall that $\Phi$ is the distance tensor where the value at the indices $i, h, w$ is the squared L2 distance between prototype $i$ and each location $(h, w)$ in the feature map.

$$\mathcal{L}_{proto-feature} = \mathcal{L}_{cell} + \mathcal{L}_{bg} \tag{4}$$

$$\mathcal{L}_{cell} = \frac{1}{K_{cell}} \sum_{i=1}^{K_{cell}} \Phi_{i,h_{max},w_{max}}; \quad \mathcal{L}_{bg} = \frac{1}{K_{bg}} \sum_{i=K_{cell}+1}^{K} \Phi_{i,h_{min},w_{min}} \tag{5}$$

where $K_{cell}$ and $K_{bg}$ are the number of cell prototypes and background prototypes, respectively. In Eq. 5, we use $(h_{max}, w_{max})$ indicating the location of the maximum density in the ground truth density map for the cell prototypes. For the background loss component in Eq. 5, $(h_{min}, w_{min})$ indicates the location of the minimum density in the ground truth density map. We introduce this new prototype-to-feature loss term to encourage cell prototypes to be similar to features in regions containing cells and background prototypes to be similar to features in regions with no cells.

**Proposed Diversity Loss ($\mathcal{L}_{\mathbf{diversity}}$)** We introduce this loss term to encourage distinctiveness among prototypes within the same group (e.g., cell prototypes or background prototypes). By penalizing excessive similarity between prototypes belonging to the same group, this loss helps the model learn diverse and meaningful patterns, which are crucial for capturing the variability in cellular and background regions. The loss is defined as:

$$\mathcal{L}_{\text{diversity}} = \frac{1}{2} \sum_{g \in \{\text{cell,bg}\}} \frac{1}{K_g(K_g - 1)} \sum_{i,j} \mathbf{L}_g[i, j], \tag{6}$$

where $\mathbf{L}_g$ is the masked similarity matrix for each group $g \in \{\text{cell}, \text{bg}\}$, representing cell and background groups, respectively. This ensures that diversity is enforced separately for prototypes representing cell regions and those representing background regions. To compute $\mathbf{L}_g$, we first define the matrix for the prototypes for each group $g$ as $\mathbf{P}_g \in \mathbb{R}^{K_g \times d}$, where $d$ is the feature dimension and $K_g$ is the number of prototypes for group $g \in \{\text{cell}, \text{bg}\}$. Using $\mathbf{P}_g$, the cosine similarity matrix for prototypes within each group is computed as:

$$\mathbf{Q}_g = \mathbf{P}_g \mathbf{P}_g^T. \tag{7}$$

To penalize only excessive similarity, an element-wise threshold function is applied:

$$\mathbf{Z}_g = \max(0, \mathbf{Q}_g - \tau_g \mathbf{1}), \tag{8}$$

where $\tau_g$ is the similarity threshold for group $g$, and $\mathbf{1}$ is a matrix of ones. To exclude self-similarity, the diagonal elements of $\mathbf{Z}_g$ are masked using the identity matrix $\mathbf{I} \in \mathbb{R}^{K_g \times K_g}$:

$$\mathbf{L}_g = \mathbf{Z}_g \odot (1 - \mathbf{I}), \tag{9}$$

where $\odot$ denotes element-wise multiplication. By focusing only on off-diagonal similarities that exceed the threshold, $\mathcal{L}_{\text{diversity}}$ ensures that prototypes within each group (cell or background) are distinct and diverse. This separation of diversity enforcement by group allows the model to capture nuanced patterns specific to cellular and background regions, enhancing both interpretability and robustness.

**Summary.** CountXplain can handle a wide range of cell counts with fewer prototypes via the proposed loss terms instead of requiring prototype labeling. Prototype labeling makes INSightR-Net (Hesse and Namburete, 2022) and ExPeRT (Hesse et al., 2024) impractical for cell counting tasks with very diverse cell counts. Additionally, CountXplain provides background prototypes, offering insights into non-cellular artifacts.

## 3. Experiments and Results

**Datasets.** We used two public cell counting datasets: **IDCIA** (Mohammed et al., 2023) and **DCC** (Marsden et al., 2018). We chose all 119 DAPI-stained fluorescence microscopy images from IDCIA with an average of $141 \pm 120$ cells per image. These images pose challenges such as blurry regions, clustering of cells, lighting variations, and high cell counts. We used 100 images for training and the rest for testing for this dataset. DCC contains 176 images of various cell types, averaging $34 \pm 22$ cells per image and offering diverse experimental conditions and cell densities. We follow the split size used by (Guo et al., 2019) with 100 images for training and 76 for testing for DCC. For both datasets, ground-truth density maps were generated by applying Gaussian blurring to the expert-annotated cell locations.

**Implementation.** CountXplain was implemented using PyTorch and PyTorch Lightning. We used the original code of CSRNet and ExPeRT. CSRNet, was trained for 200 epochs with a batch size of 1 to accommodate varying input sizes (Li et al., 2018). We varied hyperparameter values and selected the ones offering the lowest Mean Absolute Error (MAE) for each dataset. The selected CSRNet model was trained with the learning rates of $1e-6$ for DCC and $7e-5$ for IDCIA. We set $K$ to 6 for DCC ($K_{cell} = K_{bg} = 3$) and 8 for IDCIA ($K_{cell} = K_{bg} = 4$), each with a prototype depth $d$ of 64. Loss weights were 1 for $\lambda_1$ and $\lambda_2$, and 100 for $\lambda_3$, with a similarity threshold $\tau_g = 0.8$ for both cell and background prototypes. When training CountXplain, we used batch sizes of 32 and 16 for DCC and IDCIA, respectively, and a learning rate of 0.01. Following Chen et al. (Chen et al., 2019), prototype projection was performed every 100 epochs so that each prototype could be visualized using the corresponding image from the training set. For ExPeRT, we used 6 prototypes on DCC and 8 on IDCIA. See Appendix A for more details.

### 3.1. Results

**Counting performance.** Table 1 demonstrates that CountXplain can maintain a similar Mean Absolute Error (MAE) to CSRNet while adding interpretability. This is notable

Table 1: Mean and std. of MAEs of cell counting on each test dataset from 5 trials.

| DME | Self-interpretable | Method | ↓ DCC | ↓ IDCIA |
|:---:|:---:|:---:|:---:|:---:|
| ✓ | ✗ | CSRNet (Li et al., 2018) | $2.61 \pm 0.27$ | $3.49 \pm 0.68$ |
| ✓ | ✗ | SAUNet (Guo et al., 2019) | $3.0 \pm 0.3^*$ | - |
| ✗ | ✓ | ExPeRT (Hesse et al., 2024) | $4.63 \pm 2.19$ | $37.24 \pm 15.28^{\dagger}$ |
| ✓ | ✓ | CountXplain (ours) | $2.59 \pm 0.23$ | $3.42 \pm 0.40$ |

*MAE reported by the original work; $^{\dagger}$See Appendix A for discussion

given the challenging characteristics of the datasets, including large image size, overlapping cells, and diverse imaging conditions. ExPeRT, a self-interpretable regression model, trades accuracy for interpretability, yielding a higher MAE than CSRNet. CountXplain, on the other hand, has a lower MAE compared to ExPeRT by at least 2.04 (mean difference). INSightR-Net (Hesse and Namburete, 2022) was omitted as it is suited for ordinal regression tasks, not counting tasks. See Appendix A for more information.

These quantitative results are particularly encouraging as they demonstrate that our focus on interpretability does not come at the cost of accuracy. This achievement addresses a common concern in interpretable machine learning, where increased transparency often leads to a trade-off in performance.

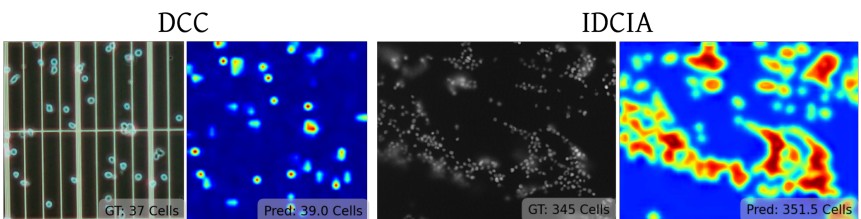

Figure 2: Examples of predicted density maps by CountXplain

Figure 2 shows the robustness of CountXplain in effectively handling sparse cell distributions in the DCC dataset while accurately capturing the dense and crowded cell regions in IDCIA. ExPeRT, though self-interpretable, cannot display spatial distributions of cells.

**Models' Global Knowledge:** To identify global patches, we compute a similarity map between each prototype and the training images, extract the local bounding box with the highest similarity, and select the top three patches based on descending similarity scores. Figure 3 illustrates the differences in cell and background patterns recognized by CountXplain, providing biologists with a rough understanding before applying the model to their dataset. Each row in Figure 3 presents the three most similar image patches for each model's prototype.

**Preliminary Expert Survey for Prototype Understanding.** Thus far, there is no consensus on how to evaluate interpretation methods (Molnar, 2022). Since machine learning interpretation is rather new to this domain, we conducted a preliminary survey with biologists to gain insight into whether the prototype groups effectively captured their intended characteristics (cells or non-cell artifacts).

**Survey Design:** Three cell-counting experts evaluated 24 images containing red-boxed regions (identified by thresholding similarity maps at the 99th percentile and connected components algorithm (Virtanen et al., 2020)) to create local bounding boxes. Experts

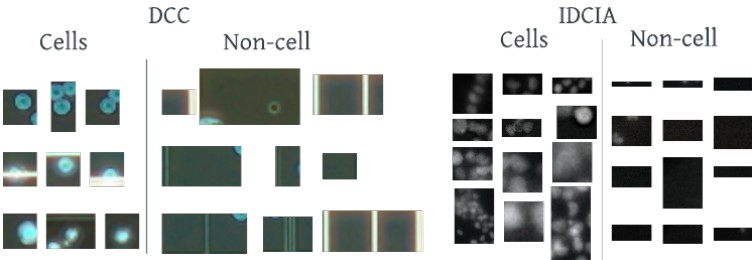

Figure 3: Global knowledge of patterns recognized by individual CountXplain models

classified what the red boxes indicated in each image as *Cell* or *Non-cell artifact* and rated classification difficulty (3: Easy to 1: Hard). Appendix C has examples of survey questions. Each participating expert has more than 2 years of experience analyzing microscopy images, specifically counting cells.

**Findings:** The evaluation highlighted the strong interpretability of our prototype groups. **Cell prototypes:** The identification agreement was high, with a mean of 97.25% and perfect consensus achieved in 91.7% of the samples. The ease of classification for cell prototypes was similarly robust, with a mean rating of 2.69 (SD = 0.37). However, one sample, rated lowest with 67% agreement and a mean ease score of 1.67, indicated a need for refinement in certain cases. **Background prototypes:** The results were even more consistent. The identification agreement matched that of cell prototypes, averaging 97.25%, with 91.7% of samples achieving perfect consensus. All background prototypes were rated as *easy* to classify, yielding a mean ease score of 3.00 (SD = 0). This uniform clarity underscores the reliability of the background prototypes in distinguishing non-cell artifacts.

These findings demonstrate the effectiveness of our prototype groups in representing their respective features. Cell prototypes showed strong interpretability, with only minor limitations in specific cases, while background prototypes consistently excelled in clarity and agreement. Together, these results validate the utility of our method for interpretable density map-based cell counting.

**Ablation study.** Table 2 demonstrates that removing diversity loss significantly decreases the distance between prototypes of the same group, indicating collapse and a failure to capture varied feature patterns. In contrast, including diversity loss increases these distances, suggesting that the prototypes effectively learn diverse and representative features.

Table 2: Effect of diversity loss on intra-class distances in preventing prototype collapse.

| | | Minimum | | Average | |
|---|---|---|---|---|---|
| Other loss | Diversity Loss | Cell | Background | Cell | Background |
| ✓ | ✗ | 0.69 | 0.01 | 1.09 | 0.01 |
| ✓ | ✓ | 1.66 | 1.85 | 2.01 | 1.85 |

**Impact of number of prototypes (K):** Figure 4 illustrates the effect of varying the number of prototypes $K$ on MAE for the DCC and IDCIA datasets. The results indicate that $K = 6$ for DCC and $K = 8$ for IDCIA yield the lowest MAE, suggesting that these values provide an optimal balance for accurate counting. Notably, when $K = 2$, the performance is suboptimal since the model lacks sufficient prototypes to capture variations in cell appear-

ances, and the diversity loss cannot be applied due to the presence of only one prototype per group.

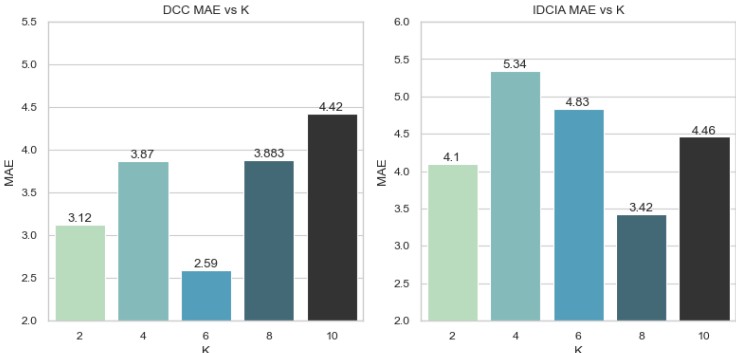

Figure 4: Analysis on the impact of the value of $K$ on MAE. Lower MAEs are prefered.

**Impact of the prototype-to-feature loss:** While CountXplain still achieved a Mean Absolute Error (MAE) of 2.60 in counting performance, Figure 5 reveals a critical issue: the absence of clear differentiation between cell and background (non-cell) prototypes. As highlighted by the red boxes in the non-cell column, multiple background prototypes visually resemble cell prototypes despite representing non-cell regions. This visual similarity between supposedly distinct prototype categories demonstrates that the prototype-to-feature loss function serves as an essential control mechanism, ensuring that cell and background prototypes learn distinct and relevant features. Without this targeted guidance, background prototypes incorrectly capture cell-like features instead of learning true background characteristics, undermining the model's interpretability.

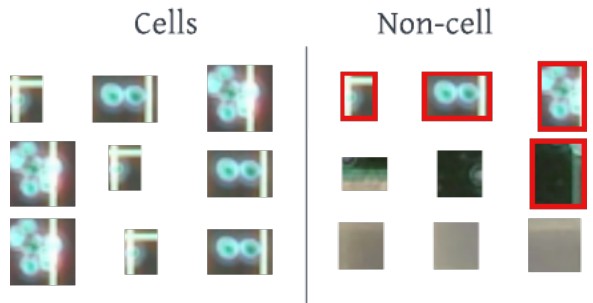

Figure 5: Model's global knowledge when the prototype to feature loss is not used.

## 4. Conclusion and Future Work

In this work, we introduced CountXplain, an interpretable framework for cell counting using prototype-based density map estimation. Our approach matches state-of-the-art effectiveness while providing transparent decision-making through prototype-based explanations. Experiments demonstrate its robustness across diverse cell appearances and imaging conditions. Future work will explore larger user studies and dynamic prototype allocation based on image complexity.

## Acknowledgments

This work is partially supported by National Science Foundation Award No. DGE 2152117. Findings, opinions, and conclusions expressed in this paper do not necessarily reflect the view of the funding agency. We thank Prof. Surya Mallapragada for providing additional domain background.

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

## Appendix A.  Additional Details

**Choices of Datasets**: We chose the DCC and IDCIA datasets to illustrate that our CountXplain does not sacrifice predictive performance while providing interpretability. Previous work has shown tradeoffs between accuracy and interpretability for self-interpretable methods (Luo et al., 2019; Wang et al., 2021). ProtoVAE (Gautam et al., 2022) is a self-interpretable method for image classification tasks that show statistically no reduction in accuracy on datasets with small image sizes of up to 32x32. Image sizes in biomedical datasets can be much larger, making it challenging to develop a self-interpretable method without sacrificing predictive performance. The DCC image size is between $306 \times 322$ and $798 \times 788$ while the IDCIA image size is 800x800. The image sizes in the chosen datasets are larger than those of the existing datasets for cell counting. Other datasets for cell counting have lower image sizes from 60x60 to 600x600 (Mohammed et al., 2023; Paul Cohen et al., 2017).

**Training:** Following (Zhang et al., 2016), we obtained the ground truth density maps by placing a Gaussian kernel at each dot annotation representing individual cells. Images in the DCC and IDCIA datasets were resized to 256x256. The training used a momentum of 0.95, a weight decay of $5 \times 10^{-4}$, and the Adam optimizer (Kingma and Ba, 2015). The feature extractor was based on a pre-trained CSRNet (Li et al., 2018) model, with weights frozen to preserve its pre-trained knowledge. For prototype-based training in CountXplain, we used a batch size of 32 and a learning rate of $1e^{-2}$. The training was conducted for 500 epochs or until convergence, based on validation performance. Similar to CSRNet (Li et al., 2018), the predicted density map size of CountXplain is $\frac{1}{8}$ of the input image size. No image augmentation was used in CountXplain. Following (Guo et al., 2019), for each trial, models were trained on a fixed random training split and tested to compute MAE. We repeated this five times, reporting the mean and variance (except ExPeRT on IDCIA).

In the ExPeRT model's approach, at inference time, predictions are made using a weighted average of prototype labels within a certain radius $r$. If none of the distances of the prototypes are within a given $r$, then the model would not be able to make predictions. For DCC, we used an $r$ value of 5. On the other hand, we experimented with different values for $r$ on IDCIA as the minimum distances to each prototype are much larger. Hence, $r$ ranging from 5 to 200 were tested and the minimum MAE for each trial was taken. Finally, we presented the mean and std of those values in Table 1.

## Appendix B. Additional ablation studies

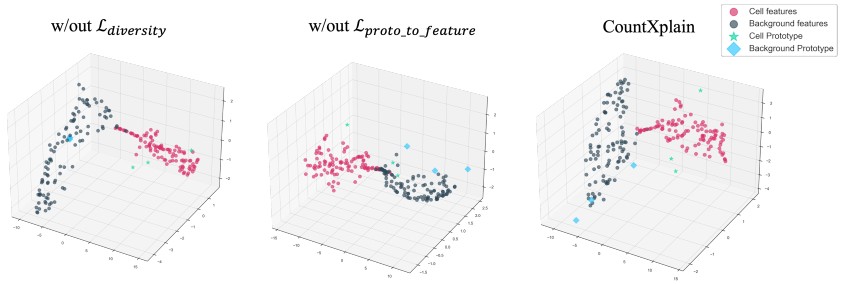

Figure 6: Qualitative ablation result on impact of loss components

To evaluate the effectiveness of CountXplain's components, we reported our qualitative assessment of the impact of the diversity loss ($\mathcal{L}_{diversity}$) and prototype-to-feature loss ($\mathcal{L}_{proto\_to\_feature}$). Figure 6 shows 3D PCA visualizations of the learned prototypes and their corresponding features under different loss configurations. Without the diversity loss (left), the background prototypes (diamonds) exhibit severe collapse, clustering in a limited region of the feature space despite the spread of background features (gray). When removing the prototype-to-feature loss (middle), we observe a significant misalignment between prototypes and their corresponding feature distributions - both cell prototypes (stars) and background prototypes (diamonds) are positioned away from their respective feature clusters (pink and gray).

In contrast, CountXplain's complete loss function (right) achieves both prototype diversity and proper feature alignment. The background prototypes are well-distributed across the background feature space, while cell prototypes effectively anchor different regions of the cell feature distribution. This demonstrates how the diversity loss prevents prototype collapse while the prototype-to-feature loss ensures meaningful relationships between prototypes and their corresponding features, enabling robust cell detection and counting.

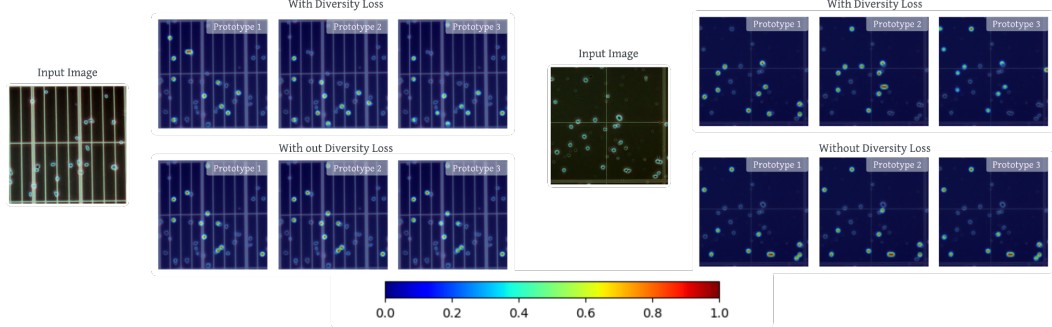

Figure 7: Similarity maps for each prototype with and without the diversity loss

Additionally, Figure 7 supports the importance of the diversity loss in the training of CountXplain. The figure presents a qualitative comparison of prototype activations with and without diversity loss. When diversity loss is included, the model learns prototypes

that activate on distinct regions of the image, capturing a broader range of variations in cell appearances. In contrast, without diversity loss, the closest patches to different prototypes tend to overlap significantly, indicating that the model learns redundant representations. This supports our claim that diversity loss encourages the prototypes to capture different aspects of the data distribution, improving interpretability while maintaining counting performance.

## Appendix C. Expert Evaluations

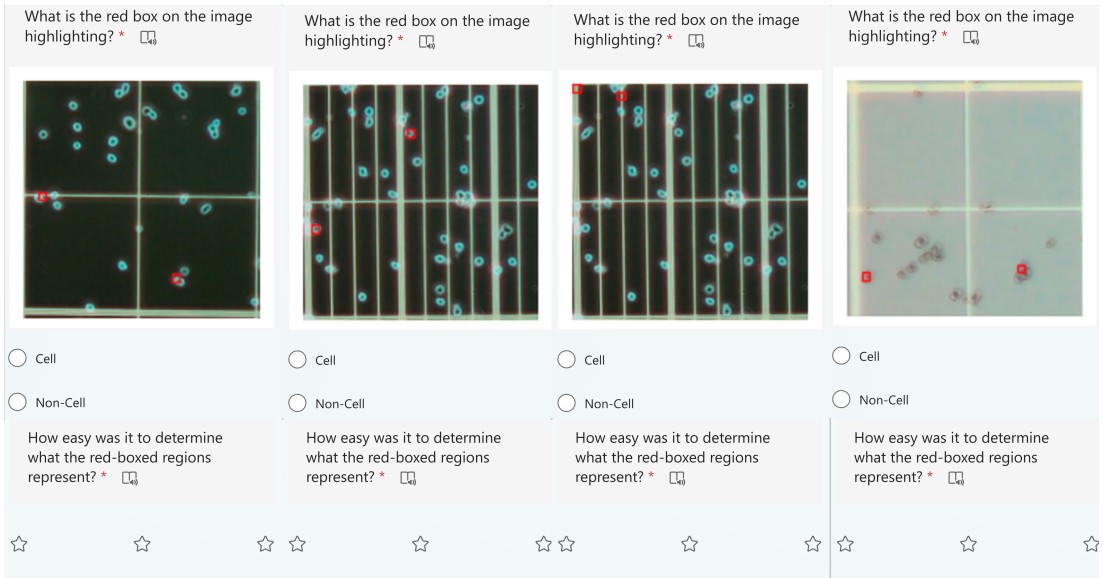

Figure 8: Survey question examples to domain experts. Bounding boxes are created using connected components algorithm after thresholding with the 99th percentile.

We conducted an online survey. Each expert was asked two questions for each image. (1) What is the red box on the image highlighting? and (2) How easy was it to determine what the red-boxed regions represent? See Figure 8. Each red box indicates an image patch covering a region matching prototypes with 99 percentile similarity. We do not show red boxes on every cell in the image since we have density maps to show spatial distributions of cells, as shown in Figure 10.

**Individual Explanations:** While the prototypes provide a global understanding of the model, they are also used to explain individual predictions. Specifically, the similarity of each prototype is combined with the weights of a convolutional layer, which are learned during training, to provide explanations for individual samples. For a given location on a predicted density map, the explanation for the corresponding prediction is computed as:

$$\hat{D}(\mathbf{x}) = \sum_{i=1}^{K} \theta_i \cdot S_i(\mathbf{x}) \tag{10}$$

where $S_k(\mathbf{x})$ represents the similarity between the input sample $\mathbf{x}$ and the $k$-th prototype, and $w_k$ are the learned weights for each prototype. These weights indicate the contribution of each prototype to the final prediction, allowing us to provide an explanation that is specific to each individual sample.

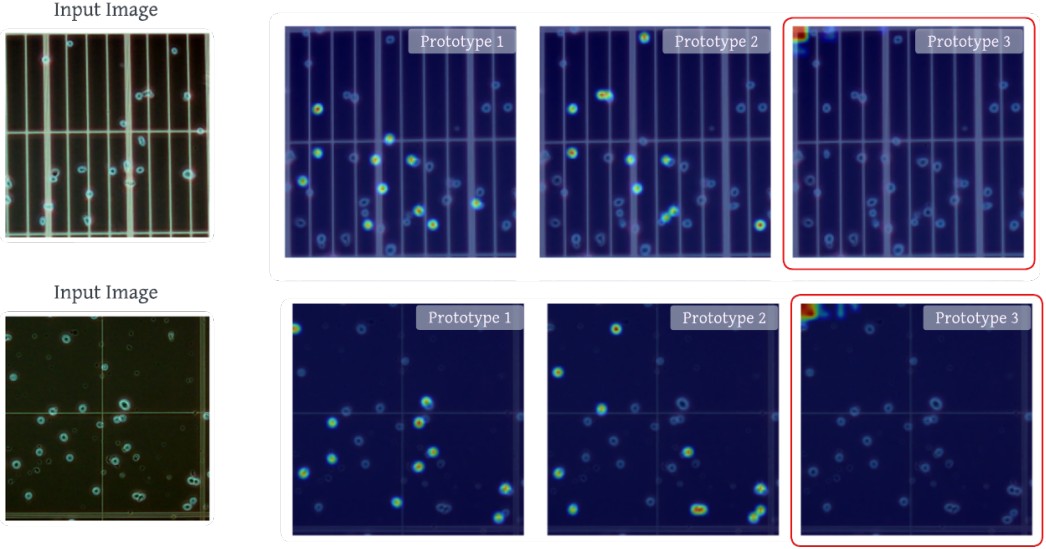

Figure 9: CountXPlain's sensitivity to $\tau$ value on the DCC dataset. When $\tau = 0$ Prototype 3 fails to capture prototypes relevant to cells.

Figure 9 shows the sensitivity of CountXPlain to the similarity threshold $\tau$ on the DCC dataset. When $\tau = 0$, the model strictly enforces diversity, preventing prototypes from capturing subtle variations in cell patterns. This is evident in Prototype 3, where the model fails to capture meaningful cell features and instead activates on background regions. Allowing a small degree of similarity with $\tau > 0$ enables the model to capture more relevant cell features across prototypes, improving interpretability.

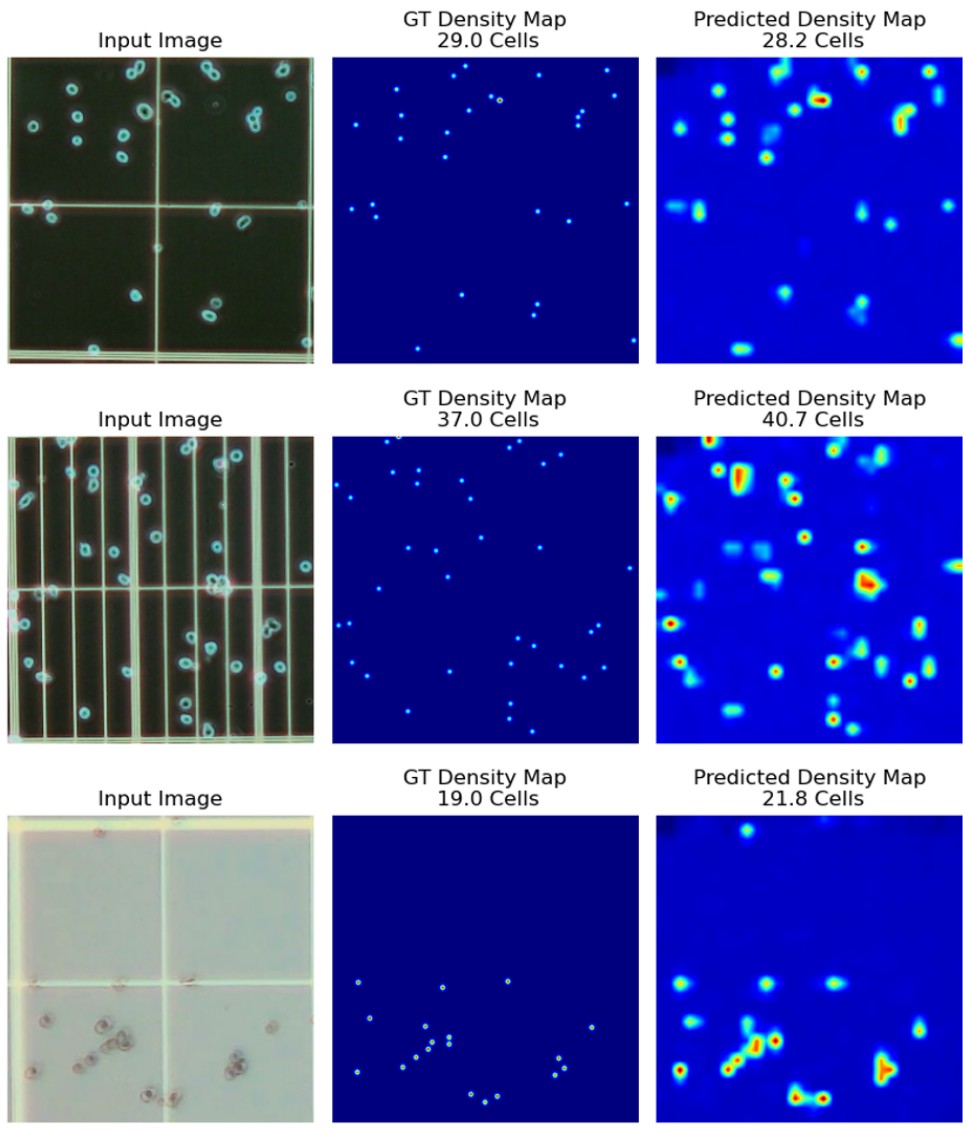

Figure 10: CountXplain's predicted density maps corresponding to the images in the survey examples.

