# OpenReview forum: "CountXplain: Interpretable Cell Counting with Prototype-Based Density Map Estimation"
_MIDL.io/2025/Conference — MIDL 2025 Oral_

### Official Review · Reviewer_uR11 · 2025-02-20

**Confidence:** 3
**Preliminary Rating:** 4
**Recommendation:** Poster
**Final Rating:** 4

**Summary:**

This paper proposes a novel prototype-based cell counting method that improves the interpretability of density map estimation for cell counting. The authors integrate a prototype layer into the DME framework and train the model to simultaneously learn prototypes of cells and background while performing density estimation. The experiments on two datasets show that the proposed method archives high accuracy while enhancing the transparency of the algorithm. The authors performed an expert survey to evaluate the interpretability of the prototypes.

**Strengths:**

This paper addresses an important topic and proposes a novel approach to improve the interpretability of the cell counting model.
Experiment results demonstrated the effectiveness of the proposed method.
The motivations for designing such prototype-based interpretable model are well explained.
This paper is well-written. Both the methods and experimental results are explained very clearly.

**Weaknesses:**

(1) It seems that the prototypes are only used to provide a global explanation for the model. They are not used to explain individual samples, which limits the contributions of the proposed method.
(2) From Figure 3, the selected patches for the same prototype vary a lot, making interpreting these prototypes difficult.
(3) Since the number of prototypes K affects the interpretability and different values of K were used for different datasets in the experiments, the authors should provide more details on how K affects both performance and interpretability. Additionally, they should clarify why they did not simply use two prototypes—one for cells and one for the background—and explain the criteria used for selecting the number of prototypes.

**Detailed Comments:**

In section 2.1, the authors state: “We choose CNN since labeled datasets in this domain are relatively much smaller than those in generic domains”. I don’t understand why CNN can solve the issue caused by small datasets.

**Justification Of The Final Rating:**

I appreciate the authors' rebuttal. My questions have been clarified and addressed in the paper. The proposed method is interesting, and I believe the paper is well-structured and has the potential to contribute to the community.

**Justification Of The Preliminary Rating:**

This paper proposes a novel model that enhances the interpretability of density map estimation for cell counting. The experiments show that the model can achieve good performance while being more transparent. However, some key details are not thoroughly discussed, and the interpretability of the prototypes remains somewhat unclear.

**Questions To Address In The Rebuttal:**

Please address the points in “weaknesses” and “detailed comments”.

---

> ### Author Response · Authors · 2025-03-08
> **Rebuttal for Reviewer uR11**
>
> We appreciate the reviewer's insightful feedback on our paper.
>
>
> > Individual explanations
>
> While the prototypes provide a global understanding of the model, they can also be used to explain individual predictions. Specifically, the similarity of each prototype is combined with the weights of a convolutional layer, which are learned during training, to provide explanations for individual samples.
>
> For a given location on a predicted density map, the explanation for the corresponding prediction is computed as:
>
> $\hat{D}(\mathbf{x}) = \sum_{i=1}^K \theta_i \cdot S_i(\mathbf{x})$
>
> where $S_k(\mathbf{x})$ represents the similarity between the input sample $\mathbf{x}$ and the $ k $-th prototype, and  $w_k$ are the learned weights for each prototype. These weights indicate the contribution of each prototype to the final prediction, allowing us to provide an explanation that is specific to each individual sample.
>
> > Selection criteria for the value of $K$
>
> The number of prototypes $K$ was selected based on counting performance. Using only two prototypes — one for cells and one for the background — would not capture the diverse visual appearances of cells, such as occluded cells, cells on hemocytometer lines, and isolated cells. Increasing $K$ allows the model to learn these variations. Figure 4 in the revised version shows the MAE of CountXPlain for different $K$ values, where $K=6$ for DCC and $K=8$ for IDCIA provided optimal performance.
>
> > Why CNN solves the problem with small datasets
>
> The statement in question was intended to justify our choice of Convolutional Neural Networks (CNNs) over Vision Transformers (ViTs) for this task. Specifically, CNNs are known to perform better on smaller datasets due to their inherent local inductive biases, which help in learning meaningful representations efficiently.

---

> > ### Comment · Area_Chair_Tu9Y · 2025-03-14
> >
> > Dear Reviewer,
> > does this address your comments?
> > Your AC

---

### Official Review · Reviewer_PZ2j · 2025-02-21

**Confidence:** 4
**Preliminary Rating:** 4
**Recommendation:** Poster
**Final Rating:** 5

**Summary:**

The authors adapt ProtPNet, a prototype based interpretable-by-design deep learning model, for the task of density map estimation and cell counting. They propose two new loss terms to encourage prototypes to be similar to their respective class (background/cell) and for prototypes of the same class to be diverse. Using two public datasets (IDCIA and DCC), they demonstrate that their model, CountXplain, can achieve a similar performance to standard neural network architectures, while providing additional interpretability. In terms of measuring interpretability, the authors conducted a preliminary study asking three experts to label whether the prototypes referred to cells or background and found that the experts overwhelmingly agreed with the model’s assignments.

**Strengths:**

- The paper is clear and well written
- The authors propose a novel application of part-protoype networks
- The authors propose useful adjustments to the ProtoPNet training/architecture for density map estimation
- Expert evaluation of the model
- Evaluation on multiple public cell-counting datasets

**Weaknesses:**

Table 2 is not enough evidence to show that the proposed diversity loss increases the diversity of the prototypes
- The primary question I have here is “Are there qualitative differences between the prototypes?” Table 2 and figure 4 indicate there are clear differences in the model activations of the prototypes, but is there a qualitative difference in the features of the image? The authors should additionally provide a qualitative comparison of the prototypes the model learns with/without the diversity loss so that the difference in prototypes can be seen.
- Additionally, it is unclear why this loss is required. If the model requires more diverse prototypes to successfully estimate the density map then should the density loss component not be sufficient? If the argument is that by enforcing diversity, even when the training set does not require it, increases generalisation, then it would be desirable to both state and demonstrate that.


There is insufficient ablation/sensitivity analysis. I appreciate some additional ablation in the appendix but..
- Why is K set to 6 for DCC and 8 for IDCIA? How sensitive is the model to these values? The diversity loss is added to encourage diverse prototypes, but with only 3 prototypes per class for DCC, how diverse can they be?
- How sensitive is the diversity loss to $\tau_g$? And why was 0.8 chosen?
- Appendix A shows how the PCA visualisations of the learnt prototypes change when the diversity loss and/or prototype to feature loss is removed, but how is the MAE affected in each case?

More evidence is required to demonstrate an improvement upon previous work
- Why did you choose 6 prototypes on DCC and 8 on IDICIA for ExPeRT, when the original paper uses 100 prototypes? If the method is designed for more prototypes to be used, it feels like an unfair comparison to use the hyperparameters that work for CountXplain. The choice in hyperparameters should be justified and evidence that ExPeRT does not perform better with more prototypes should be provided.

**Detailed Comments:**

I don’t understand your references to ExPeRT requiring prototype labels but your method not requiring them. My understanding is that both methods specify whether prototypes refer to the cells or the background (or in ExPeRT’s original application, the relevant gestational age), so either way prototype labels are used. Can you explain this further?

There is a typo near the bottom of page 4: “interoperability” should be “interpretability”.

Although the specific form of the proposed diversity loss is novel, Djoumessi et al. [1] proposed a similar loss "dissimilarity loss" to encourage diverse prototypes.

[1] Djoumessi, Kerol, Bubacarr Bah, Laura Kühlewein, Philipp Berens, and Lisa Koch. ‘This Actually Looks like That: Proto-BagNets for Local and Global Interpretability-by-Design’. MICCAI 2024. https://doi.org/10.48550/arXiv.2406.15168.

**Justification Of The Final Rating:**

Most of my questions have been answered satisfactorily so I have increased my score to strong accept. I still think the paper would be improved by more thorough comparisons to previous work (e.g. Reporting MAEs for two different K values in a rebuttal -- not even in the main paper -- is not an adequate demonstration that the optimal hyperparameters for ExPeRT have been found). However, the majority of my concerns have been addressed and the proposed method is clearly of interest to the community and deserved of acceptance.

**Justification Of The Preliminary Rating:**

Overall, the paper is well written and the application and methodology justified, but in the paper’s current state it is not entirely clear which aspects of the proposed method contribute most and the hyperparameter choice for comparisons to the literature is not well justified.

The authors have adapted a well known method and successfully applied it to a novel domain and while there is nothing technically wrong with the paper there is substantial room for improvement, hence my recommendation for weak accept.

**Questions To Address In The Rebuttal:**

All weaknesses mentioned earlier. However, the most important two are:
- The weakness in the comparison to ExPeRT
- The lack of thorough ablation and sensitivity analysis

Overall, the paper is well written and the application and methodology justified, but these two improvements would make readers much more confident that the method is better than previous work and make readers understand the effect of each aspect of the proposed method.

---

> ### Author Response · Authors · 2025-03-08
> **Rebuttal for Reviewer PZ2j (1/2)**
>
> We appreciate the reviewer's detailed feedback on our paper.
>
> > Are there qualitative differences between the prototypes when diversity loss is not used?
>
> To answer this question, we have included visual examples comparing the prototypes learned with and without the diversity loss See Figure 7 in the revised paper). Specifically, in the DCC dataset, we observe that when the diversity loss is not used, the closest patches to all prototypes tend to correspond to the same image regions. This suggests a lack of diversity in the learned prototypes, reinforcing the importance of the diversity loss.
>
> > The need for diversity loss
>
> The density loss component is primarily responsible for ensuring that the predicted density maps align with the ground truth density maps. However, it does not explicitly encourage diversity in the learned prototypes. Without the diversity loss, we observe that prototypes tend to collapse, meaning that multiple prototypes focus on similar or even identical image regions, leading to redundant representations.
>
> The diversity loss addresses this issue by ensuring that different prototypes capture distinct feature representations, which enhances interpretability and improves generalization by encouraging the model to learn a richer set of discriminative features. This is particularly supported by the visual examples in the appendix, which show that the diversity loss helps the model to learn more diverse and meaningful prototypes.
>
> > Sensitivity to K
>
> The value of $K$ is set empirically based on the performance of the model on the respective datasets. For the DCC dataset, we observe that $K = 6$ and for IDCIA,  $K = 8$ provide optimal performance in terms of counting accuracy. This choice is based on the results of our experiments, where we evaluated the counting performance of CountXplain with $ K $ values ranging from 2 to 10. These results demonstrate that the model performs best with these values for both datasets. We have included these results in the revised manuscript (Figure 6) to provide more insight into the choice of $K$.
>
> Regarding the diversity of prototypes, we recognize that cells share common visual characteristics, which naturally leads to some level of similarity between prototypes. However, there are still subtle differences in how cells appear in the images, such as occlusion by lines, cells appearing in groups, and cells located individually. The diversity loss is designed to capture these subtle differences and encourage distinct prototypes, even with only 3 prototypes per class in DCC. Although the prototypes may not be drastically different, the diversity loss helps ensure that they represent a broader spectrum of the variations present in the data.
>
> > How sensitive is the diversity loss to $\tau_g$? And why was 0.8 chosen?
>
> We chose the value of $\tau_g = 0.8$ as it provided better interpretability compared to setting $\tau_g = 0$. When $\tau_g = 0$, the model enforces strict diversity among prototypes, which may prevent it from capturing subtle variations in cell appearances. As shown in Figure 9 of the revised version, this can lead to some prototypes (e.g., Prototype 3) failing to capture relevant cell features and instead focusing on background regions. Allowing a small degree of similarity ($\tau_g = 0.8$) helps the model capture better cell patterns, resulting in more meaningful prototypes and improved interpretability.
>
> > The diversity loss and/or prototype to feature loss is removed, but how is the MAE affected in each case?
>
> When the diversity loss is removed, CountXplain performs with an MAE of 3.01 on the DCC dataset and 4.96 on the IDCIA dataset. In contrast, when the diversity loss is included, CountXplain achieves an MAE of 2.59 on DCC and 3.42 on IDCIA. These results demonstrate that the inclusion of the diversity loss improves the model’s performance by reducing the MAE on both datasets.

---

> > ### Author Response · Authors · 2025-03-08
> > **Rebuttal for Reviewer PZ2j (2/2)**
> >
> > > Number of prototypes for ExPeRT
> >
> > The decision to use 6 prototypes for the DCC dataset and 8 for the IDCIA dataset was based on empirical results. We also experimented with using 100 prototypes for ExPeRT, as done in the original ExPeRT paper, but observed a significant drop in performance. Specifically, the MAE increased to 23.94 on DCC and 37.44 on IDCIA. This suggests that increasing the number of prototypes does not improve ExPeRT's performance on these datasets.
> >
> > > Prototype labels in ExPeRT
> >
> > ExPeRT assigns continuous labels to each prototype, which are uniformly sampled from the label range of the training dataset. These labels are used to guide the model’s predictions, and the labels remain fixed throughout the training process. The assignment of uniform labels in ExPeRT is appropriate for tasks where the prototype labels correspond to some continuous value (e.g., gestational age) and where all prototypes are treated equally, regardless of whether they correspond to distinct groups like cells or backgrounds.
> >
> > In contrast, our method does not require assigning continuous labels to prototypes. This is because, in cell counting tasks, the dataset is often long-tailed (few images with a high number of cells and a high number of images with a low number of cells), and assigning uniform labels to prototypes may not be ideal. Instead, we have two distinct groups of prototypes: cell and background without additional continuous labels.
> >
> >
> > > 'dissimilarity loss' to encourage diverse prototypes.”
> >
> > Our diversity loss is calculated separately for cell and background prototypes, with a goal of encouraging diversity between cell prototypes and between background prototypes. Additionally, to account for the subtle similarities that naturally exist between cells, we introduce a parameter$\tau_g$, which allows some level of similarity between prototypes, specifically penalizing only excessive similarity. This ensures that the diversity loss does not overly penalize prototypes that share some inherent visual characteristics (e.g., generally common features), but still encourages meaningful diversity where appropriate.
> >
> > Thank you for pointing out the similarity to Djoumessi et al.'s work. We would be happy to include it under related works.

---

> > > ### Comment · Area_Chair_Tu9Y · 2025-03-14
> > >
> > > Dear Reviewer,
> > > does this address your comments?
> > > Your AC

---

### Official Review · Reviewer_Vgz8 · 2025-02-26

**Confidence:** 4
**Preliminary Rating:** 4
**Recommendation:** Poster
**Final Rating:** 4

**Summary:**

This work proposes a novel architecture that combines a prototype-based approach with density map estimation to provide accurate cell counts as well as reasoning of the model’s output. Its architecture ensures that the model inherently learns prototypes that represent global patterns of cell and non-cell types in the data. This is done by defining the final density map, from which the counts are computed, as a weighted combination of similarities between each location in the image and each prototype. The combination of losses ensures accurate counting / density as well as diversity among prototypes. It ensures that cell prototypes represent cells and background prototypes represent background regions.

**Strengths:**

1.	The method is novel and well supported by the conducted experiments on two datasets.
2.	The baselines are well chosen.
3.	It includes a survey with biologists that confirm the learned prototypes are meaningful, i.e. cell prototypes clearly show cells and background prototypes clearly show background.
4.	The writing is clear and coherent.
5.	The work seems to be reproducible with code availability and a clear explanation of architecture, implementation and training.

**Weaknesses:**

1.	The prototypes were created in a way such that they would not need labelling due to the nature of the cell counting scenario and varying number of cells per image. The learned prototypes can distinguish between cell and non-cell, however, they do not show a meaningful clustering of patterns per prototype. The prototypes are diverse, yes, but I do not see significant differences between patterns of each prototype, especially for the non-cell patterns. Maybe the number of prototypes is unnecessarily high for the types of images? (see Fig. 3)
2.	119 Images were (manually?) chosen out of the 262-image sized IDCIA dataset. It is unclear how and why a subset was created.
3.	The datasets used for training and evaluation are rather small with only 119 and 176 images included respectively. However, I am not familiar with these types of datasets so this might be the norm due to immense labelling effort required.

**Detailed Comments:**

-	2.1 Model Architecture: The first paragraph is unclear and not coherent with the following paragraphs. Please provide clarity on what exactly is meant by “based on the Convolutional Neural Network (CNN) architecture”. I am guessing the authors refer to the pre-trained feature extractor in the CSRNet from Li et al. (2018) . Further, I am guessing the authors explain why they chose this feature extractor which is pre-trained due to the limited availability of labelled datasets for cell counting and this is what they mean by “We chose CNN since labelled datasets in this domain are relatively smaller than those in generic domains.” Please provide more clarity here and include that the pre-trained feature extractor is from the CSRNet which is currently only stated in the appendix.
-	3.1 Results: Models’ global knowledge: “Each row in Figure 3 shows three image patches with the highest similarity with each model’s prototype.” This sentence lacks clarity without having read the appendix.

**Justification Of The Final Rating:**

The authors of this paper address an important research gap - namely, the interpretability of cell counting methods for microscopy images. They propose a novel approach that takes a step toward filling this gap while still providing accurate counts, as clearly supported by their experiments.

I do remain unconvinced of the necessity of having so many prototypes, as they do not exhibit distinct groupings. That said, the additional experiment justifying the choice of $K$ effectively demonstrates that using only two prototypes - one for cells and one for the background - is insufficient for accurate counting. I would be interested in seeing future work exploring these prototypes further. Perhaps two diverse prototypes could be sufficient if the architecture were adjusted to ensure that reducing the number of prototypes does not compromise counting performance.

**Justification Of The Preliminary Rating:**

The authors propose a novel method which claims accurate cell counting and meaningful prototypes as a form of interpretability. These claims are clearly supported by the conducted experiments. My main concern is the lack of clear separation between patterns clustered by each prototype.

**Questions To Address In The Rebuttal:**

Please address the two comments under "Detailed Comments". Comment 1 under "Weaknesses" might also be part of future work if justified.

---

> ### Author Response · Authors · 2025-03-08
> **Rebuttal for Reviewer Vgz8**
>
> We want to thank the reviewer for their thorough feedback on our paper.
>
> > Clarity on what exactly is meant by “based on the Convolutional Neural Network (CNN) architecture”
>
> The statement in question was intended to justify our choice of Convolutional Neural Networks (CNNs) over Vision Transformers (ViTs) for this task. Specifically, CNNs are known to perform better on smaller datasets due to their inherent local inductive biases, which help in learning meaningful representations efficiently. This aligns with findings in prior works such as Raghu et al. [1], Park et al. [2], and d'Ascoli et al. [3], which demonstrate that ViTs often underperform on small datasets unless architectural modifications are made to incorporate local inductive biases.
>
> > Pre-trained CSRNet backbone
>
> Yes, we use the pre-trained backbone of CSRNet \cite{li_csrnet_2018}. We have revised the text to state that the feature extractor is based on CSRNet explicitly.
>
> > Further clarification on Figure 3
>
> To improve clarity, we have moved the relevant explanation from the appendix to the main paper in the revised version.
>
> > Not enough difference between prototypes
>
> One key challenge in this problem setting is the lack of concept-labeled data, which prevents direct supervised training of prototypes with predefined categories. Instead, our approach allows the prototypes to emerge naturally from the data distribution, with the diversity loss ensuring that different types of cells and background regions are represented.
>
> While the learned prototypes effectively distinguish between cell and non-cell regions, the non-cell prototypes primarily capture blank spaces or hemocytometer lines, as illustrated in Fig. 3. Since most background regions in the dataset consist of blank areas, the diversity among background prototypes may appear less pronounced. However, the inclusion of multiple background prototypes still ensures that variations, such as the presence or absence of hemocytometer lines, are accounted for. This helps domain experts to know what type of background (non-cell artifacts) the model has learned.
>
> [1] Raghu, Maithra, et al. "Do vision transformers see like convolutional neural networks?." Advances in neural information processing systems 34 (2021): 12116-12128.
>
> [2] Park, Namuk, and Songkuk Kim. "HOW DO VISION TRANSFORMERS WORK?." 10th International Conference on Learning Representations, ICLR 2022. 2022.
>
> [3] d’Ascoli, Stéphane, et al. "Convit: Improving vision transformers with soft convolutional inductive biases." International conference on machine learning. PMLR, 2021.

---

> > ### Comment · Reviewer_Vgz8 · 2025-03-10
> >
> > I want to thank the authors for their clarifications regarding my comments. I also highly appreciate the additional included figures and text passages further explaining the choice of $K$ and the influence of the diversity loss (appendix).

---

### Official Review · Reviewer_y2j5 · 2025-02-26

**Confidence:** 4
**Preliminary Rating:** 4
**Final Rating:** 4

**Summary:**

The paper investigates the task of cell counting in microscopy images. For this task, density map estimation and cell counting seem to be SOTA methods. However, there is a gap to fill in terms of explainability for these methods. In this paper, authors proposed a new strategy based on the addition of a prototype layer, associated with a prototype loss and a diversity loss to ensure that the model handles diversity in images. The proposed method is compared to three SOTA methods on two publicly available datasets. Results show comparable performance to non-interpretable SOTA methods while adding crucial information: explainability. A survey of clinicians is performed to judge the correctness of the prototypes, and results seem quite correlated with ground truth.

**Strengths:**

The paper proposes an interesting strategy, focusing on improving existing methods with interpretability, clearly filling a gap for density map estimation. Results are promising and validated by clinicians, which is particularly important when working on healthcare applications. Code is publicly available, which is highly appreciated.

**Weaknesses:**

There is no major comments. Ablation study shows only the influence of diversity loss, but it would be great to see the influence of different prototypes e.g. inclusion of background, etc. Visual comparison of density maps of ExPerT would have been appreciated.

**Detailed Comments:**

No major comment. Maybe a more detailed ablation study, exploring the effect of different parametrisation of the loss would have been useful to better understand which component helps the most.

**Justification Of The Final Rating:**

I would like to thank the authors for submitting their rebuttal and answering the reviewers' comments.
The new figures and sentences better explain the choice of K and the influence of the prototype-to-feature loss.
As the authors mention in their comment, ExPerT is also "self-interpretable", so it would have been interesting to compare the explanations provided by this model compared to the proposed CountXplain, especially for clinical practitioners that have been surveyed here, since the addition of interpretability is the main contribution of this work.

**Justification Of The Preliminary Rating:**

The paper is well-written, comprehensive and it proposes a novel strategy for a highly impactful topic. While the analysis could have been strengthened with more breakdown metrics, I think this paper is of interest to the community and fills a gap in biomedical imaging by providing interpretable results on the task of cell counting using density maps. I recommend acceptance.

**Questions To Address In The Rebuttal:**

The paper is very well written, comprehensive and proposes a novel strategy for a highly impactful topic. I recommend acceptance.
Visual comparison of density maps of ExPerT would have been appreciated.

**Special Issue:**

No

---

> ### Author Response · Authors · 2025-03-07
> **Rebuttal for Reviewer y2j5**
>
> We appreciate the reviewer's feedback and suggestions.
> > Visual comparison of density maps of ExPerT would have been appreciated.
>
> Regarding the visual comparison of density maps, we would like to clarify that ExPeRT does not generate density maps, as it is a regression model. Its output is a scalar value rather than a spatial representation. We included ExPeRT in our study because it provides interpretability in the context of regression tasks, offering insights into how the model makes predictions.

---

> > ### Comment · Area_Chair_Tu9Y · 2025-03-14
> >
> > Dear Reviewer,
> > does this address your comments?
> > Your AC

---

### Author Rebuttal · Authors · 2025-03-08

**Rebuttal:**

We thank the reviewers for their thoughtful feedback and constructive suggestions. We have carefully addressed each of the reviewers' comments and provided detailed responses, along with corresponding changes to the paper. To facilitate easy reference, all modified text is in purple. We believe these changes have strengthened our paper, and we hope it solves the weaknesses mentioned.

**Supporting Material:**

/attachment/2ed7176f95b812be5ecab8acfe3cc54167562107.pdf

---

### Meta-Review · Area_Chair_Tu9Y · 2025-03-19

**Recommendation:** Accept (Poster)
**Confidence:** 4

**Metareview:**

The authors develop a prototype-based architecture for cell counting in microscopy images, ensuring interpretability of the results by learning  prototypes of cells. The reviewers agree that the paper is well written, proposes a novel idea and provides sufficient evaluation and validation of the proposed method. All reviewers are in agreement to accept the paper (3 weak, 1 strong).